# Transforming Intensive Animal Production: Challenges and Opportunities for Farm Animal Welfare in the European Union

**DOI:** 10.3390/ani12162086

**Published:** 2022-08-15

**Authors:** Mariann Molnár

**Affiliations:** Animal Welfare Program, Faculty of Land and Food Systems, University of British Columbia, 2357 Main Mall, Vancouver, BC V6T 1Z4, Canada; mariann.sz.molnar@ubc.ca

**Keywords:** animal welfare, farming, confinement, reform, transformative change

## Abstract

**Simple Summary:**

The European Union (EU) has made commitments to review all established policies on farm animal welfare and by the end of 2023 propose new regulations that will improve farming conditions and phase out the use of cage-based systems. A case study of pig farming in Hungary suggests that farmers are much constrained in their ability to make such significant changes to their farming operations; hence, a purely legislative reform may not succeed in delivering the desired changes. Competing socio-economic interests and constraints created by global trade regulations require that reformers extend their efforts from the limited approach of policy review to addressing issues that are well beyond the control of farmers. These include economics-, legislation-, and technology-induced concerns. To ensure a truly effective transition, reforms need to address those factors that push production toward the use of confinement systems.

**Abstract:**

Since the 1960s, the European Union (EU) has made efforts to ensure the welfare of farm animals. The system of EU minimum standards has contributed to improved conditions; however, it has not been able to address the deeper factors that lead to the intensification of animal farming and the consolidation of the processing sector. These issues, along with major competitive pressures and imbalances in economic power, have led to a conflict of interest between animal industries, reformers, and regulators. While the priorities of the European Green Deal and the End the Cage Age initiatives are to induce a rapid phasing out of large-scale cage-based farming systems, the industry faces the need to operate on a highly competitive global market. Animal farmers are also under pressure to decrease input costs, severely limiting their ability to put positive animal-care values into practice. To ensure a truly effective transition, efforts need to go beyond new regulations on farm animal welfare and address drivers that push production toward a level of confinement and cost-cutting. Given the right socio-economic and policy incentives, a transition away from intensive farming methods could be facilitated by incentives supporting farm diversification, alternative technologies, and marketing strategies.

## 1. Introduction

In the current European Union (EU) farm animal welfare reform effort, there is wide agreement that complex, interconnected factors affect farm animal welfare. To date, farm animal welfare advocacy approaches have recognized that farmers and the general public have conflicting norms, that there are inconsistencies between societal principles and consumer purchasing behaviors, and that the uniform enforcement of existing legislative standards is problematic. However, to deliver meaningful change to farm animals, additional issues also require careful attention. These include ensuring good-quality animal care by sufficient numbers of well-trained staff [1] and addressing the constraints on farmers caused by the economic power imbalance between farmers and the highly consolidated meat processing companies that buy their products [2,3]. Farm animal welfare reforms also need to address the actual impact of the current legislative tools [3] and how the norms of farmers and their ability to transform their farming operations vary with the size and type of farming methods they currently use [4].

My understanding of these issues began with a research project into confinement and alternative pig farming in Hungary, where, unlike much of the EU, the transition to intensive confinement production is still in progress [5]. The research showed some of the constraints faced by farmers, including “technological lock-in” after farmers have invested heavily in confinement systems, and the external pressures on farmers caused by low and fluctuating profits.

This paper briefly reviews the evolution of the European pro-welfare reforms up to the recent publication of the EU Green Deal, the Farm to Fork Initiative, and the End the Cage Age citizen initiative and then assesses the shortcomings of the principally legislative approach that the EU has adopted. The paper then assesses efforts, challenges, and opportunities for transformative change in animal agriculture in light of the Hungarian case study.

## 2. From Silent Spring to EU Pro-Welfare Legislation and Back Again

Industrial methods of food production were first problematized by Rachel Carlson (1962) in the book *Silent Spring* [6], which critiqued the abundant use of DDT and other pesticides in intensive crop production in the USA. This book was soon followed by Ruth Harrison’s (1964) account of emerging large-scale indoor animal agricultural practices in the UK and their effects on the health and welfare of farm animals [7]. In a foreword that she wrote to Harrison’s book, Carlson advocated a Schweitzerian ethic based on reverence for life [8] and asked: “how can [farm] animals produced under such conditions be safe or acceptable human food?”

Although ethical concern for animals in the UK dates back to at least the 18th century [9], reforms in the 20th century rested on emerging scientific evidence, the activities of professional non-governmental organizations, and a growing recognition of animal sentience. The UK’s early anti-cruelty law dating back to 1822 was supplemented in 1968 by legislation specifically targeting farm animal welfare in intensive farming systems. Provisions in this Agriculture Miscellaneous Provisions Act resulted from a complex of ethical, economic, and public concerns for animals [10], followed in 1979 by the UK Farm Animal Welfare Council (FAWC) publishing the well-known Five Freedoms [11], which have provided high-level guidance for reform.

With concern for farm animal welfare spreading beyond the UK, in 1976, the Council of Europe published a convention on “the protection of animals kept for farming purposes” [12], opening a new phase of farm animal welfare reform efforts. Indeed, while the 1957 Treaty of Rome still identified animals as goods [13], the 1979 Council of Europe Convention was the first major European legislation that aimed to protect the welfare of farm animals raised in confinement systems. Subsequently, the Treaty of Amsterdam (1997, later affirmed by the 2007 Treaty of Lisbon) recognized animals as sentient beings [14]. In 1998, the European Union adopted the 1976 Council of Europe Convention and passed Council Directive 98/58/EC, concerning the protection of animals kept for farming purposes [15].

These legal documents, along with the so-called Protocol on Animal Welfare under the Amsterdam Treaty, established the ruling principles “in key areas of European law and policy making” [16] (p. 197). In addition to the general requirements laid down in the directive on issues such as appropriate staffing of animal holdings, inspection, record keeping, and automatic and mechanical equipment, minimum standards were prescribed on freedom of movement; buildings and accommodation; animals kept outdoors; feed, water, and other substances; mutilations; and breeding procedures [15]. In addition, specialized minimum standards were established on slaughter and killing practices (1993), laying hens (1999), transport conditions (2005), chicken kept for meat production (2007), calves (2008), and pigs (2008).

Therefore, since the 1960s, the EU has made major efforts to safeguard farm animal welfare by defining minimum standards [16]. These rules, however, principally target the actions of farmers by defining basic standards of care and aspects of on-farm housing, but they tend to ignore other players in the value chain that may have a major influence on the decisions of farmers and the welfare of farm animals. Moreover, although Member States are at liberty to pass national rules that extend beyond the established EU legal framework, norms higher than common EU standards have rarely been adopted.

Although these legislative reforms have brought many benefits, such as obligatory pre-slaughter stunning, serious problems remain. While the Council of Europe identifies animal welfare as an issue of common cultural heritage and acknowledges duty to care [13], the level of protection to be delivered to farm animals in many cases seems open to interpretation [14,17]. A critical examination of the principally legislative approach to ensure farm animal welfare in the EU by means of enforcing minimum standards indicates some of the weaknesses of the present system. High on the legislative agenda are issues such as animal health and food safety [18,19], while low on the agenda are issues such as further increasing comfort standards or reducing boredom for farm animals. Other concerns that are not addressed include the responsible consumption of animal products, overproduction [20], and food loss and waste [21]. Many would argue that the intensification of agricultural production; the consolidation of animal faming and processing industries; and the widespread use of large-scale, indoor confinement farming methods are key areas where change is most needed [22]. However, in practice, current EU policies do nothing to ensure a transition away from these methods. Standardized legislation also fails to account for the diversity of farm scales, technologies, and management practices that directly influence day-to-day animal welfare conditions [1,3].

These challenges are now being acknowledged in the EU by the European Green Deal [23,24] and the Farm to Fork Initiative [25], which present a much more complex understanding of the range of issues that need to be tackled. Within the context of sustainable development, the strategies propose to deliver significant improvements to farming conditions, including a substantial increase in the welfare of farmed animals, and to assess the impact of farming technologies used. The momentum for change also seems to be present in the social arena, as 1.4 million EU citizens have recently signed the End the Cage Age initiative launched by Compassion in World Farming [26], which calls on the EU to phase out the use of confinement farming methods. Key goals of the current EU agriculture reform, therefore, are to review the EU’s entire policy system for agriculture and reflect on environmental (sustainability, environmental health, biodiversity), animal health (food security), and animal welfare issues along the entire food chain, with a special emphasis on removing farm animals from confinement systems. This new approach is proposed to “foresee better living and transport conditions, and enhanced protection of animals during slaughter” [27]. It is scheduled to be presented by the end of 2023 after extensive consultation [26].

In summary, despite the substantial farm animal welfare reforms since Carlson and Harrison, intensive farming methods are again in the spotlight. Further transformative change in animal agriculture may be possible only if, in addition to legal protection, other important aspects, such as the socio-economic context of animal farming and ethical principles guiding decision-making, are also given sufficient attention [28].

## 3. Understanding the Need for Change

Effective EU policies on agriculture, food production and farm animal welfare are thought to be challenged by a set of competing interests that could result in compromised arrangements [29]. Competing interests (human vs. animal welfare) and the contradictory mandates (animal welfare vs. economic interests) of the EU may be suggested to induce significant challenges to the animal welfare reform effort. Depending on the context of policy analysis, some would argue that the EU farm animal welfare reform effort to date was able to act against “industrial” forms of animal agriculture and represents a “counter-commoditization strategy” [19] (p. 77), delivering major animal welfare improvements [30]. However, critics of the current legislative approach claim that pro-welfare legislation only focuses on “the irrational property owner” who inflicts harm on animals without reasonable human benefit [31] (p. 187) and highlight the difficulty of an ever-increasing legislative burden on farmers [14]. These views are supported by individual farmers or farmer associations that express their concerns over the tension between EU regulations and production efficiency on a global market [32].

This tension between EU regulations and global trade in food has been understood as the outcome of World Trade Organization (WTO) rules of conduct [19]. WTO trade regulations necessitate the application of neo-liberal free trade principles [19] offering limited chances for a consistent set of rules governing both trade interests and pro-welfare action. As Hobbs et al. [33] note, under WTO regulations, the EU is obliged to facilitate the international trade of “like” products that cannot be distinguished from one another. Obligatory labeling of the production method is a clear farm animal welfare interest but cannot be carried out due to concerns over market distortion [33,34,35].

While free trade in its present form continues to be widely supported, a diverse set of data indicate that damage or harm caused by large, intensive industries cannot be sufficiently addressed on the market [28,36,37]. A dominant assumption of free trade is that it is beneficial to consumers, who make choices consistently with and according to their individual values [38]. This would mean that growing concern for and awareness of farm animal welfare could—in theory—empower consumers to relieve animals from harmful production methods [39]. Most willingness-to-pay studies emphasize this premise [40], yet other studies suggest that direct causality between public concern and consumer action cannot be found [41,42], as important self-protective mechanisms, such as cognitive dissonance [43], the complexity of the market [44], or price sensitivity [40], could prevent consistent purchasing behaviors. Falk and Szech [42] also find that moral decisions may be affected when participants feel that they have no direct influence on an immoral act and therefore exhibit “a tendency to lower moral values, relative to individually stated preferences” (p. 710). Findings therefore suggest that moral concern alone is unable to counter-balance negative market externalities [42], including harm caused to animals.

Consistent animal advocacy is also needed. Since the 1960s, societal concern for the welfare of farm animals has evolved, and European animal welfare advocacy approaches have changed over time. Prior to the publication of *Animal Machines* [7], the UK was already debating the problem of making farming “more profitable, modern and efficient” [45] (p. 17), leading to concerns over farm animal stress, suffering, and cruelty [45,46]. After the publication of Harrison’s book, farm animal welfare, a “fundamentally new language and concept” [45] (p. 14), emerged from a wide-ranging scientific, political, and public debate [46]. Following recommendations by the Brambell Committee, animal welfare developed into an established scientific discipline [47,48] and advocacy approach [31]. Most pro-welfare policies were developed on the animal welfare premises, acknowledging the subjective state of welfare matters related to animals and that given the possibility, they are ready to contribute to their own individual state of well-being, including the achievement of pleasure and the avoidance of pain [48,49,50,51,52,53]. A welfarist perspective is therefore predominantly concerned with the quality of an animals’ life rather than its quantity (unless longevity is used as an indicator of welfare) and has further developed to ensure “good lives” for animals under human care irrespective of their use or value [54]. An animal welfare approach therefore focuses on establishing and ensuring ethical human–animal interactions via legislation, education, capacity building, and good practice [55].

However, other ethical frameworks, challenging the animal welfare approach, have also been proposed. Most notable of these are the consequentialist, utilitarian animal liberationist ethic developed by Singer [56] and the categorical animal rights theory proposed by Regan [57], further developed into what is now called the abolitionist theory proposed by Francione [58,59]. These theories rest on strong foundational principles and propose the application of “non-interference rights” [54] in the belief that human–animal interactions, such as farming animals and the consumption of food from animal origin, are inherently wrong [56,57,59]. Therefore, in its most extreme form, non-interference rights prescribe the total abolition of all human–animal interaction [57,58,59], while in other cases, only harmful interactions are to be abandoned for all animals [60] or those species that are comparable to humans in their capacity to suffer [56]. The animal welfare approach and its competing categorical hegemonic discourses and ethical frameworks [61,62] lead to the fragmentation of scientific, social, and political reform efforts [28] solely based on moral disagreement. These ethical debates have not been sufficiently resolved since the 1960s [45], and evidence suggests that an increasing number of animal advocacy approaches are based on non-interference rights in the US as well as the EU [63,64].

The model developed by Anderson [28] predicts that powerless groups, such as children or animals, gain protection only if there is a common ethical foundation and a clear understanding of what needs to be done. Hence, success in the EU reform efforts will likely depend on establishing a clear, united ethical imperative guiding reformers on what needs to be done for animals and why [28]. To date, the current, ongoing EU farm animal welfare reform effort has not presented such a comprehensive approach that could take reform beyond the review of existing policies. Some argue that “it will not be easy to reach a consensus on what animal welfare is and how it should be achieved/improved” [65] (p. 116). However, it is likely that without attempting to broaden the reform effort, the present status quo of animal farming will prevail and animal welfare reforms may lead to progress on only a limited subset of issues [66].

In the related field of environmental protection, alternative economic models seem to be developing. A circular economy, for example, presents a united moral and innovative economic framework that seems to already function within the established neo-liberal free trade context [67]. For farm animal welfare, continuous efforts provide progress in establishing a functional moral imperative [68]. Such examples include *Reverence for Life*, by Schweitzer [8]; Biosocial Communitarianism by Callicott [69,70]; the One Health initiative [71]; and A “Practical” Ethic for Animals by Fraser [54]. Anderson [28] finds that “once a new ethic is firmly established… it can be… as powerful as legal reform… Without this ethical shift, in fact, mere legislative reform will probably be ineffective” (p. 62).

## 4. Challenges for Transformative Change in Animal Agriculture

While it may seem that farmers can freely decide on how they keep their animals, Molnár and Fraser [3] found in a case study of pig farming in Hungary that producing for the ever-changing mainstream market severely constrains farmers’ choices. With the large-scale consolidation of the processing industry, a large number of farmers compete to sell almost identical (generic) products to a relatively small number of processors [3,72]. Larger, better-established companies thus have greater access to processors, leading to competitive advantages over smaller companies that sell fewer animals [73]. Competitive pressures on animal producers also seem to be induced by the inability of farmers to negotiate the price of their finished animals. As the price of pigs and other intensively reared animals usually goes in cycles [23,24], farmers either opt to sell to processors based on a long-term contract, setting a fixed price, or sell based on the daily rate determined by the market [3]. In both cases, farmers will have to endure significant periods of losses [2,22] and will therefore have to make savings during more profitable times to allow them to continue to produce [3]. Free trade also enables international sourcing; hence, for example, if local feed prices increase, processors can import animals from a distance and local farmers will be unable to recover the cost of production [3,73], again leading to further economic vulnerabilities of farmers.

With this imbalance in economic power between farmers and the processing industry, the rational adjustment of product supply and demand would be necessary. However, Molnár and Fraser [3] discovered that well-definable features of animal farming prevent producers from adapting to challenges induced by the market. Unlike other industries, animal farming cannot respond quickly enough to lower prices or demand because production can only be adjusted after a significant time-lag (in the case of pig farming, at least 6 months or more) and in these periods, the market might change significantly [3]. This inability of the animal farming sector to increase or decrease production as the market dictates also induces a pressure on farmers to either further intensify or abandon their farming operations, a feature that is already widely apparent in industrialized countries [2,22,74]. Consequently, those farmers who would like to remain competitive have to ensure a high level of production efficiency. This need for efficiency seems to be the most important reason why farmers who produce for the mainstream market build and then continue to use production methods that necessitate large-scale, automatized confinement systems [3].

Most intensive confinement farming operations also specialize in producing a single commodity and invest in costly technologies that are suited for only a single species. These housing technologies have been found to be difficult to adjust. Technological lock-in not only seems to prevent farmers from being able to enjoy the economic resilience that a mixed-farming enterprise would be able to offer [3] but also predisposes the farmers to the continued use of these large, fixed investments [75,76]. The economic pressures that force farmers to decrease input costs also lead to decreased amenities for animals, including space, care staff, and labor costs, which further limit the possibility for farmers to put positive animal-care values into practice [2,3,72]. In addition, while much research has emphasized that farmers using confinement methods are mostly concerned about animal health and productivity [77,78,79,80,81], this “entrepreneurial discourse” [2] was found to represent the beliefs of only the largest, most intensive farmers (≥1000 sow operations) [4]. Farmers operating medium-scale enterprises (≈400–600 sows), who used the same methods, but at a lower scale, were more critical of intensive confinement systems, and in ideal circumstances, would have provided their animals with increased welfare through higher standards of care and more natural living conditions [4]. Yet, even these famers struggled to transition their technologies and increase farm animal welfare standards [82]. Therefore, the combined effects of intensive farming technologies and economic pressures significantly constrain the actions of farmers, despite their values [4].

Challenges endured by farmers in the EU and beyond have significantly impacted the scale and intensity of farming operations [2], the health and well-being of farmers [83,84], the countryside and rural communities [85], and overall farm animal welfare conditions [12]. Yet, current standard (top-down) solutions seem to address only a limited subset of the issues. In terms of regulatory approaches, inconsistent enforcement presents significant challenges [13]. Additional problems have also been induced by the narrow focus of policies on certain inputs that may not necessarily ensure good welfare outcomes (e.g., flooring and ventilation), the neglect of other determinants that may lead to poor welfare (frequent group mixing, boredom, etc.), and the unpredictability of events on a farm that can limit compliance (e.g., unusually high birth rates) at certain times [3]. The economic status of farmers seems to also limit the effectiveness of legislation, especially if farmers face prolonged periods of losses and eventually lose so much that the level of care given to animals is compromised [3]. Although generally well appreciated by farmers, the subsidy system was also seen to be compromised. At times when the prices of finished animals were low, subsidies were found only to prevent major losses [3], while at the time of high prices, subsidies were not enough to allow major welfare-related investments [3,82]. Subsidies also seemed to be designed for confinement farms and therefore may have indirectly incentivized a move to intensive systems [3]. Finally, a reliance on consumers exercising informed choice did not seem to apply well to the mainstream market as processors were found to exhibit a preference for uniform “commodity” production, which without consistent labeling prevented the differentiation of products [3].

Following global trends, the consolidation of animal agriculture is still ongoing in the EU [3] and beyond. This process seems to facilitate the continued expansion of already successful operations, while others are forced to abandon production due to the inability of farmers to compete [3,5] or their unwillingness to grow the scale of their farming operations beyond a certain size [4,5]. Challenges associated with confinement farming induced by an uneven distribution of economic pressures and power seem to affect those medium-scale farmers who, given the chance, may be willing to engage in farming practices more aligned to their values [4]. A lack of transformative change in production methods may therefore lead to a further increase in the scale and distribution of intensive practices [2,5].

## 5. Opportunities for Transformative Change in Animal Agriculture

Despite the challenges, history indicates that as in the case of the successful European child labor reforms that liberated children from mines and factories, change is possible to ensure the welfare of animals [28]. A growing number of perspectives are calling for large-scale intensive confinement agriculture to be transformed into a better model that ensures good lives for farm animals [86], the responsible use of natural resources [87], the conservation of biodiversity and wildlife habitats [88], and the production of safe and wholesome food, while ensuring food security [89] and improving food distribution [90]. While this paper primarily argues for the case of farm animal welfare, these goals appear of equal importance and seem to depend on one another. While the dominant form of agricultural production in the developed world is intensive confinement agriculture, alternative methods of production and retail present important models worth exploring.

Alternative agricultural production is a collective term representing a highly diverse set of animal farming methods [91], yet in comparison to confinement farming methods, key characteristic features identified in the case of pig farming are that these systems are generally small-scale extensive or semi-intensive farms with a high level of personal involvement of the farmer, where animals are kept in loose group housing with access to indoor shelters and outdoor runs and with longer (at least double) life spans of both breeding and fattening animals [4,5]. These farms are based on a low-capital model, as they do not invest in and rely on expensive automated technologies and are therefore free of most technological constraints [3]. Alternative farming also seems to rely on diversification (mixed farming methods) to ensure resilience during periods of fluctuating prices and costs [3,5]. While production efficiency is lower and variable costs of production are higher than on confinement farms, alternative farming operations possess a greater level of stability as farmers rarely sell their finished animals to processors but pursue niche markets, often engaging directly in the processing and sale of their products [3,5]. As UNIDO [92] also finds, a shorter food chain can secure a higher return for the farmer as well as other benefits to rural communities, biodiversity, and the natural environment.

A transition away from intensive confinement farming methods is sometimes assumed to compromise food security for a growing world population due to the inefficiency of alternative farming systems [93] and the need for more land [94] and agricultural workers [95]. Intensive confinement production seems to provide a solution to these important challenges, but only when examined without full cost-accounting of harms caused [96] and in the absence of assessing the combined negative effects of overproduction [20], food loss and waste [97], and the effect of automation on worker wages [98]. Indeed, important factors driving intensification in animal farming seem to be inherently induced by the replacement of human labor with automated technologies, first to save on wages and more recently to make up for the lack of motivated workers for such facilities. The ever-increasing rate of urbanization is taking its toll on rural communities [99], and a number of complex factors influence the willingness of people to engage in farming [100]. The advantages of emerging alternative farming methods, such as organic or ecological farming [101], permaculture [102], and silvopastoral systems [103], seem to be just as well understood by farmers [5] as they are accepted by both EU citizens and consumers [41] and appear to address multiple challenges induced by current agricultural practices [4,5].

A bottom–up transition away from the use of monoculture farming may enable the production of a more diverse set of products on a single unit of land, with well-adapted and more resilient plant varieties and animals being produced in more natural conditions. Diversity in farming practices, seeds, and animals can also enable the recovery of soil health and even provide an opportunity to restore biodiversity [104]. It does not inevitably mean a complete transition away from the use of “smart” technologies or some machinery. However, it does seem to necessitate the preservation of rural communities and an influx of motivated, educated, and well-paid workforces. Some data already indicate a growing momentum for a new generation of farmers, who leave their urban, non-agrarian lives to engage in agricultural production [105]. Other studies support the idea that consumers who recognize the problems of confinement systems and have direct interaction with farmers are more willing to negotiate the development of farming methods that address common concerns [106,107].

In our research, differences between qualities of life of animals in intensive versus alternative systems have even been acknowledged by some confinement farmers, especially those with medium-scale operations [4] but not by those with large operations. Hence, strategies to transition intensive production may require policies, incentives, and technological solutions that fit the priorities of producers working at different scales [4]. Large operations that are resistant to transformative change may require well-qualified staff and expertise to improve farm management [4] and technological solutions (e.g., enabling the use of bedding, roughage, and environmental enrichment) that may improve animal welfare without compromising farm efficiency or competitiveness [5]. However, small- and medium-scale confinement farmers seem most open and able to change and, given the right support, are most likely to transition their farming operation away from indoor, intensive confinement systems and incorporate more natural production methods [4]. In both cases, an increased level of transparency about farming methods will also be needed to ensure that all farm scales find societal approval and a stable market [5].

## 6. How Can Confinement Farming Be Transformed to Ensure a Meaningful Increase in Farm Animal Welfare Conditions?

The EU is highly committed to protecting the welfare of farm animals and is striving to further improve conditions. However, the EU also appears be in a difficult situation, because any transition of animal agriculture away from the use of intensive confinement methods must be balanced with other socio-economic interests. The next year will be crucial in determining the depth and breadth of EU involvement in pro-welfare action. The level of current commitments suggests that the EU will limit its reform efforts to a comprehensive review of the existing legislative system and propose a new set of regulations that will prescribe the phasing out of cage-based systems. However, these actions will likely continue to principally target farmers. In recognition of the difficulty and cost of a transition away from confinement farming methods, the EU may provide legislative and financial incentives to farmers, for example, through targeted animal welfare payments. Evidence suggests that this approach may probably lead to the largest, most intensive confinement farmers or corporations (with the most lobby power) to resist a “forced” transition [93]. Yet, given that incentives are adequate in scale and are widely available and a comprehensive agreement on the desired end point of the transition is reached, this approach may provide an opportunity for small- to medium-scale intensive confinement farmers, who already face severe economic pressures. If the EU pursues this route, European farmers will require a system of coherent protective measures to ensure that as a result of higher welfare regulations, farmers transitioning their operations will be able to sell their products and that animal production will not simply be displaced to other jurisdictions [108].

An alternative approach to the EU pro-welfare action is to go beyond existing commitments and extend the review to assess the influence of other direct and indirect factors—such as the impact of global free trade inducing a continuous pressure to increase the scale and intensity of farms—and propose new regulations and incentives that address these other important drivers of confinement farming systems. Most importantly, this new policy system will have to acknowledge problems induced by (a) the unequal distributions of power and competitive pressures on the market, (b) the economic vulnerability of farmers, (c) the high level of specialization of large-scale intensive farms, (d) the problem of technological lock-in induced by expensive confinement technologies, and (e) the inability of the farming sector to quickly adjust production to demand. The current EU animal welfare reform effort may, therefore, need to do much more than prescribe the phasing out of intensive confinement systems; it will have to challenge global agricultural production trends and provide a sound, workable alternative production and marketing strategy to the farming community. To ensure transformative change in animal agriculture and a meaningful increase in farm animal welfare conditions, farmers will also need to be involved in the process of reform [3] and be empowered to put positive animal-care values into practice [4,5].

In addition to the legislative process, opportunities for transformative change in animal agriculture should also focus on preventing the further intensification of animal farming. This could be most easily carried out in those EU Member States, such as Hungary, that are in the process of intensifying their production methods or those that still depend on small- to medium-scale semi-intensive or extensive farming systems (those found in many EU accession states). Though the task of transitioning large-scale, intensive confinement farms is challenging, inspiration for change can be found in the example of alternative production and marketing systems. Well-targeted incentives to increase farm animal welfare conditions, the ability of producers to process their own products, and a workable solution on labeling (such as the EU egg labeling scheme) may improve market access and relieve farmers of those excessive economic pressures that predominantly drive intensification.

In the ideal scenario, the current EU reform effort should facilitate a sound transition away from intensive confinement farming methods and enable the majority of animal production to be realized in small- to medium-scale, semi-intensive and extensive, diverse farming operations where animals are housed loosely and are able to access both indoor shelters and outdoor runs [5]. A new era of farming may allow significant improvement in farm animal welfare conditions as well as ensure increased resilience of farming communities. As ideals often provide a momentum for reform efforts, so it may be possible to assume that the transformation of intensive animal production is feasible [28,109,110].

## 7. Conclusions

The greatest challenge of current EU farm animal welfare reform efforts—initiated by the EU Green Deal, the Farm to Fork Initiative, and the End the Cage Age citizen initiative—is the problem of how to realize transformative change in animal agriculture when many existing pro-welfare actions compete with conflicting mandates of the EU and contradict global free trade policies. Current commitments suggest that the EU will focus on a comprehensive review of the existing legal system to propose a new set of regulatory principles and the phasing out of caged-based farming methods. However, transformative change in animal farming appears to be severely constrained by the established system of intensive confinement production and trade practices.

This paper suggests that a narrow, policy-based reform effort may increase the vulnerabilities of farmers and fail to address major forces that shape animal farming. To deliver meaningful change to farm animal welfare within and beyond the EU, reform efforts should extend the scope of legislative reform and engage in debating and devising an alternative system of animal production and trade within the current free trade context. Given a carefully devised set of legislative, economic, and market-based incentives, farmers could break away from direct and indirect pressures that drive the intensification of animal agriculture and should have the opportunity to transition away from the use of specialized large-scale intensive confinement farming methods. The pro-welfare reform effort, therefore, requires a broad, holistic approach to ensure that actions offer a realistic path for transformative change in animal agriculture and a meaningful increase in farm animal welfare standards.

## Data Availability

Not applicable.

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
