# Peer review of "Transforming Intensive Animal Production: Challenges and Opportunities for Farm Animal Welfare in the European Union"

_animals, 2022, doi:10.3390/ani12162086_

Round 1
Reviewer 1 Report
This article's topic is of great relevance and interest. The welfare of production animals closely links to the conditions of breeding and management, which, therefore, must be governed by national and international rules that protect not only the profitability of the company and public health but also animal welfare.
Honestly, however, I do not believe the article can be published in its present form. I don't think that it can be considered a case study. In my opinion, it is perhaps more like a (quite partial) review of the literature. It mainly focuses on what is happening in the EU, so it should be integrated with the description of current legislation and farming conditions in the rest of the world. Furthermore, it does not consider in sufficient depth the characteristics of small farms, which often, in terms of well-being, can be equally lacking compared to intensive farms, even with different kinds of problems.
Author Response
Re: Manuscript revision
20 July 2022.
Dear Reviewer 1,
I would hereby like to re-submit my manuscript entitled “Transforming intensive animal production: challenges and opportunities for farm animal welfare in the European Union” to the “Transition of the Food Animal Production Towards More Sustainability and Animal Wellbeing” special edition of the journal Animals.
Thank for your feedback. I have made some revisions to the manuscript and would hereby like to respond to your comments as follows:
- I agree that there is great interest in the topic and would like to point out that the special edition of this manuscript aims to contribute to a better understanding on the “hard problem” of how to transition large-scale industrial and intensive forms of animal agricultural production away from methods that may potentially harm the welfare of farm animals.
The philosophical approach of this manuscript is therefore primarily concerned with the causes of endemic and inherent problems in intensive (and not small scale) animal agriculture. It aims to explore challenges and opportunities for changing these, and – based on empirical data from previous studies – proposes that a transition to less intensive and more diverse forms of production may be able to do so. The manuscript therefore does not imply a direct positive causality between small-scale farming practices leading directly to a high level of farm animal welfare but suggests a more complex and nuanced approach to reform. It emphasises the importance of tacking the most dominant (legislative, social, economic, technological, and ethical) forces shaping animal agriculture that are rarely accounted for in such detail. It also calls for reformers to give equal attention to these issues, and act on them according to their specific (positive and negative) impacts on animal farming and farm animal welfare conditions.
- I also agree with you that farm animal welfare is closely linked to breeding and management practices, and that national and international rules may provide a certain level of protection for important interests including public health, profitability and farm animal welfare.
However, multiple authors suggest that the current, predominantly legislative approach of reform falls short to deliver meaningful change in the daily lives of farm animals. While the European Union is now engaging in a more complex reform of agriculture (see section 2. From Silent Spring to EU pro-welfare legislation and back again), farm animal welfare considerations still seem to focus only on the need for an adjustment of legislative standards in housing and care. The manuscript presents sufficient evidence to suggest that other important considerations should also be addressed, such as the impact of free trade, the interplay between farm technologies, production costs, the market value of “finished” animals, and market access for farmers, which all seem to have a substantial effect on methods of production and the ability for farmers to put positive care values into practice.
- I am also grateful that you raised the issue of the manuscript being submitted as a Case Study and would like to inform you that since the submission of the article, I was also asked by the editors of the special edition to change this to a Review type. This was approved by the Gabrielle Zhang Section Managing Director of Animals (e-mail confirmation arrived on the 20th June 2022), so I have made the necessary change in the manuscript.
- In terms of your comment on the manuscript being a “quite partial review of the literature”, I would like to argue that the paper aims to explore a wide range of issues and presents 111 citations specific to the topic, which, in combination with the case study of intensive and alternative pig farming in Hungary, enables the formation of a new theoretical understanding on how to transition farming away from intensive production methods.
I acknowledge that this thesis may present a novel way of interpreting findings, however the manuscript aims to forward a growing scholarly discourse on the farm animal reform effort and in this sense the new ideas it proposes makes it a valuable contribution to scholarly literature.
- I also agree with your observation that – in an ideal scenario - the topic would require a global outlook. However, I believe that due to the vast differences in the farming scale, breeding and housing technologies, and legislative frameworks in countries such as the US, China, Brazil or India, a global outlook may be better discussed in a dedicated peer-reviewed article.
To ensure that readers are not mislead, I have therefore narrowed down the title to specify that it addresses the transformation of animal agriculture in the EU. Finally, please also note that the text of the manuscript contains at least 53 references to the EU and Europe (e.g. in lines 8, 18, 23, 35, 54) making in clear in the wording, that the article is concerned with EU animal welfare reform-efforts only.
I hope that the minor, but meaningful changes made to the manuscript and my detailed response will be acceptable to you, and that you will find the paper fit for publication. Once again, thank you for your professional input.
Yours sincerely,
the author
Reviewer 2 Report
This is an excellent and informative paper. I don't have any particular revisions to suggest.
Author Response
Re: Manuscript revision
20 July 2022.
Dear Reviewer 2,
I would hereby like to re-submit my manuscript entitled “Transforming intensive animal production: challenges and opportunities for farm animal welfare in the European Union” to the “Transition of the Food Animal Production Towards More Sustainability and Animal Wellbeing” special edition of the journal Animals.
Thank for your very positive feedback. I would hereby like to inform you that based on the comments of Reviewer 1 I have made some revisions as follows:
- I have changed the manuscript from being a Case Study to a Review type. This was initially proposed by the editors of the special edition after submission, and was finally approved by the Gabrielle Zhang Section Managing Director of Animals (e-mail confirmation arrived on the 20th June 2022).
- I have also narrowed down the title to specify that the manuscript addresses the transformation of animal agriculture in the EU.
I hope that these minor changes made to the manuscript are acceptable to you, and that you will continue to find the paper fit for publication. Once again, I am grateful for your professional input.
Yours sincerely,
the author